# The Construction of Polycyclic Pyridones via Ring-Opening Transformations of 3-hydroxy-3,4-dihydropyrido[2,1-*c*][1,4]oxazine-1,8-diones

**DOI:** 10.3390/molecules28031285

**Published:** 2023-01-28

**Authors:** Viktoria V. Viktorova, Elena V. Steparuk, Dmitrii L. Obydennov, Vyacheslav Y. Sosnovskikh

**Affiliations:** Institute of Natural Sciences and Mathematics, Ural Federal University, 51 Lenina Ave., 620000 Ekaterinburg, Russia

**Keywords:** 4-pyridone, oxazinone, ring-opening, benzimidazole, aldehyde-lactol tautomerism, ammonium acetate

## Abstract

This work describes the synthesis of 3-hydroxy-3,4-dihydropyrido[2,1-*c*][1,4]oxazine-1,8-diones, their tautomerism, and reactivity towards binucleophiles. These molecules are novel and convenient building-blocks for the direct construction of biologically important polycyclic pyridones via an oxazinone ring-opening transformation promoted with ammonium acetate or acetic acid. In the case of *o*-phenylenediamine, partial aromatization of the obtained heterocycles proceeded to form polycyclic benzimidazole-fused pyridones (33–91%).

## 1. Introduction

4-Pyridones are important nitrogen-containing heterocycles, which have recently attracted much attention as biologically active [1,2,3,4] and natural compounds [5,6]. Polycyclic structures, such as dolutegravir, bictegravir, and cabotegravir, are used as modern inhibitors of HIV integrase for antiretroviral therapy [1,3] (Figure 1). Baloxavir marboxil also belongs to this class of these compounds and is applied as the first cap-dependent endonuclease inhibitor for the treatment of influenza [2].

The chemistry of these heterocycles is actively developed not only for the design of biologically important compounds [7,8,9], but also for effective preparation in the industry [1,2,3,10,11,12,13,14,15]. At the same time, there is a need to search for new multifarious pyridone building blocks [16,17,18,19,20,21,22,23,24,25] and convenient synthetic tools for the construction of polycyclic pyridones [1,2,3,15], including CH functionalization [26].

The general approach is well known in the literature based on the reaction of alkyl 1-(2,2-dimethoxyethyl)-4-oxo-1,4-dihydropyridine-2-carboxylates with binucleophiles to obtain the fused heterocycles [1,3,10,11,12,13,14] (Figure 1). In this case, the acid-catalyzed deprotection of the dimethyl acetal group led to 2-(4-oxopyridin-1(4*H*)-yl)acetaldehydes, which are usually considered as intermediates of polycyclic pyridone formation. Substituted 3-hydroxy-3,4-dihydropyrido[2,1-*c*][1,4]oxazine-1,8-dione was also detected as the result of hydrolysis of the carboethoxy and dimethylacetal groups as a by-product in the synthesis of dolutegravir [10]. Such structure also can be suggested as a possible intermediate for this heterocyclization [1,10]. To the best of our knowledge, only unsubstituted 3-hydroxy-3,4-dihydropyrido[2,1-*c*][1,4]oxazine-1,8-dione was prepared from comanic acid in pure form and used for the transformation with (*R*)-3-aminobutan-1-ol [3,11]. Moreover, there are data on the Ugi reaction of a pyridone-bearing aldoacid with isonitriles for the synthesis of various piperazinone-fused pyridones [27].

We decided to study 3-hydroxy-3,4-dihydropyrido[2,1-*c*][1,4]oxazine-1,8-diones in more detail in order to find new directions for the construction of polycyclic pyridones through morpholinone ring-opening reactions. These molecules bear the hidden aldehyde moiety, which can determine their high reactivity towards nucleophiles via the tautomeric equilibrium. This strategy based on the transformation with diamines can open access to new cyclic fused pyridones, which are of interest for the further design of biologically active compounds.

## 2. Results and Discussion

### Synthesis of 3-hydroxy-3,4-dihydropyrido[2,1-c][1,4]oxazine-1,8-diones **3** and Their Chemical Properties

We started with the ANRORC reaction of 5-acyl-4-pyrone-2-carboxylate **1** with 2,2-dimethoxyethylamine as the effective method for the preparation of 4-pyridones [3,17,28,29] (Figure 2, Table 1). The ring-opening transformations proceeded under reflux in toluene for 4 h to produce pyridones **2a–d,f** in 30–90% yields. Pivaloyl-substituted pyrone **1e** did not provide the desired product, and the reaction was carried in more polar MeCN, leading to pyridone **2e** in 42% yield. Compounds **2a–e** underwent the deprotection of the dimethyl acetal moiety in aqueous HCl to form 3-hydroxy-3,4-dihydropyrido[2,1-*c*][1,4]oxazine-1,8-dione*s*
**3a–e**. For 2,5-dicarbethoxy-4-pyridone **2f**, the heating in formic acid was used for the selective hydrolysis of the COOEt group at the C-2 position as a result of the promotion by the presence of the adjacent aldehyde fragment.

Pyridones **3** can undergo aldehyde-lactol tautomerism [30] and exist as acyclic aldoacid or a cyclic lactol form (3-hydroxy-3,4-dihydropyrido[2,1-*c*][1,4]oxazine-1,8-diones) (Figure 2). It is interesting to note that this type of the ring-chain tautomerism for morpholinones has not been studied before.

The ^1^H NMR spectra of products **3** in DMSO-*d*_6_ demonstrates the existence of only the lactol form. The spectral feature of the tautomer is the presence of a downfield signal of the OH group at *δ* 8.13–8.72 ppm and an ABX system of the morpholinone moiety. For the pivaloyl-substituted compound **3e**, a singlet of the methylene group and a strongly broadened singlet of the CH proton were observed probably due to the rapid interconversion between different forms.

Pyridones **3** bear the carbonyl group at the C-5 position, which can be used for further modifications of the heterocyclic fragment. Therefore, the important task included the search for selective transformations on the morpholinone fragment. We have studied the detailed influence of conditions on the ring-opening reaction of 3-hydroxy-3,4-dihydropyrido[2,1-*c*][1,4]oxazine-1,8-diones (**3b**) with 3-aminopropan-1-ol (Table 2). A mixture of methanol–toluene was used as a solvent to increase the solubility of pyridone **3b**. The reaction did not proceed without the use of catalysts even under the prolonged reflux. It was found that the transformation in the presence of acetic acid as an additive led to product **4a** in 48% yield. We suggested that the formation of 3-hydroxypropylammonium acetate occurred, which acts as a nucleophile and activator of the morpholinone moiety. However, this transformation did not proceed at room temperature, as well as with the use of the 0.2 equiv. of acetic acid.

Taking into account the effect of AcOH, we tried to use AcONH_4_ as a bifunctional catalyst [31,32] for this process. To our delight, the product was obtained in a good yield (69%) under reflux for 12 h (TLC monitoring) (Table 2). The variation of the nature of the ammonium salt or temperature did not allow for the improvement of the reaction yield.

We tried to extend the optimized conditions with the use of ammonium acetate (Method A) for other 5-acylpyridones **3** and binucleophiles for the synthesis of polycyclic 4-pyridones (Table 3). In most cases, the acyl fragment strongly influenced the reaction selectivity, and an alternative method included the use of acetic acid (Method B). Pyridones **3b,c** bearing para-substituted benzoyl fragments underwent the transformation in the presence of ammonium acetate and led to the formation of products **4a,b** in 69–75% yields. Benzoyl- and thienoyl-substituted compounds **3a,d** reacted more effectively in the conditions of method B and provided products **4c,d** in 31–52% yields. When propane-1,3-diamine was used as a binucleophile, we were not able to isolate the desired polycyclic products in a pure form directly. The precipitates that formed always contained the starting diamine in significant amounts. Next, binucleophiles bearing two carbon atoms in the linker was used for the heterocyclization. The reaction with ethylenediamine proceeded in good yields and led to the formation of imidazo[1,2-*a*]pyrido[1,2-d]pyrazine-5,7-diones **5a,b** in 78–84% yields (Method A). Thienoyl-substituted pyridone **3d** underwent the ring-opening process in the presence of acetic acid (Method B) to produce compound **5c** in 48% yield. At the same time, we failed to isolate any products in the pure form in the reaction with ethanolamine.

The peculiarity of ammonium acetate is probably associated with its solubility in a methanol–toluene mixture and the ability to promote the ring-opening process of the morpholinone ring, which leads to the formation of the aldoacid (Figure 3). Subsequent stages, including intermolecular attack of a binucleophile and intramolecular cyclization, can be catalyzed by both the ammonium cation and acetic acid. An experiment was carried out to study the reaction of ammonium acetate with pyridone **3b** under reflux. According to the ^1^H NMR spectrum of the obtained precipitate, it was found that the formation of an open-chain structure occurred (see Appendix A). Although we did not detect the aldehyde group, a singlet of the methylene group and absence of the 3-CH proton of the lactol form were observed in the ^1^H NMR spectrum.

The reaction of pyridone **3** with *o*-phenylenediamine proceeded in the presence of ammonium acetate at room temperature or under reflux and was accompanied by aromatization under the action of atmospheric oxygen (Figure 4, Table 4). The intermediate **C** was not isolated in pure form, but was detected as by-products in all cases. Carrying out the reaction under argon did not allow the selective formation of compound **C**. This result can indicate that the oxidation additionally promotes the reaction leading to the most stable product **6**.

To obtain compounds **6** in a pure form directly, the reaction was carried out at room temperature for 12 h and subsequent reflux for 2 h. In these conditions, the aromatization proceeded completely and pyridones **6** bearing the benzimidazole fragment were isolated in 33–91% yields. The reaction turned out to be sensitive to the nature of the acyl moiety, which probably determined the occurrence of side reactions. Pivaloyl pyridone **3e** led to degradation products, which did not bear the *t*-Bu group. In the ^1^H NMR spectra of compounds **6**, the downfield singlet of methylene group was observed at *δ* 5.81–5.84 ppm due to the presence of two adjacent aromatic systems.

Thus, 3-hydroxy-3,4-dihydropyrido[2,1-*c*][1,4]oxazine-1,8-diones have been synthesized and demonstrated to exist predominantly in the lactol tautomeric form. The new and convenient approach has been developed for the preparation of polycyclic pyridones based on the pyridomorpholinones via ring-opening reactions. The binucleophile linker and the nature of nucleophile centers strongly influence the reaction outcome. The most active binucleophiles in this heterocyclization process are ethylenediamine and 3-aminopropan-1-ol. It has been demonstrated that the reaction with *o*-phenylenediamine is followed by oxidation and the formation of benzimidazole-fused 4-pyridones.

## 3. Materials and Methods

NMR spectra were recorded on Bruker DRX-400 (Bruker BioSpin GmbH, Ettlingen, Germany, work frequencies: ^1^H, 400 MHz; ^13^C, 101 MHz), Bruker Avance-400 (Bruker BioSpin GmbH, Rheinstetten, Germany, work frequencies: ^1^H, 400 MHz; ^13^C, 101 MHz), Bruker Avance III-500 (Bruker BioSpin GmbH, Rheinstetten, Germany, work frequencies: ^1^H, 500 MHz; ^13^C, 126 MHz), and Bruker Avance NEO (Bruker BioSpin GmbH, Rheinstetten, Germany, work frequencies: ^1^H, 600 MHz; ^13^C, 151 MHz) spectrometers in DMSO-*d*_6_ or CDCl_3_. The chemical shifts (*δ*) are reported in ppm relative to the internal standard TMS (^1^H NMR) and residual signals of the solvents (^13^C NMR). IR spectra were recorded on a Shimadzu IRSpirit-T (Shimadzu Corp., Kyoto, Japan) spectrometer using an attenuated total reflectance (ATR) unit (FTIR mode, diamond prism); the absorbance maxima (*ν*) are reported in cm^–1^. Mass spectra (ESI-MS) were measured with a Waters Xevo QTof instrument (Waters Corp., Milford, MA, USA). Elemental analyses were performed on an automatic analyzer PerkinElmer PE 2400 (Perkin Elmer Instruments, Waltham, MA, USA). Melting points were determined using a Stuart SMP40 melting point apparatus (Bibby Scientific Ltd., Stone, Staffordshire, UK). Column chromatography was performed on silica gel (Merck 60, 70–230 mesh). All solvents that were used were dried and distilled by standard procedures. 4-Pyrones **1** were prepared according to the literature methods [28,33,34].

### 3.1. General Procedure for the Preparation of 1-(2,2-dimethoxyethyl)-4-pyridones **2**

Ethyl 5-acyl-4-oxo-4*H*-pyran-2-carboxylate **1** (0.330 mmol) was added to a cooled solution of 2,2-dimethoxyethanamine (0.0380 g, 0.361 mmol) in toluene (1 mL). The resulting mixture was stirred for 20 min at room temperature (the precipitation was observed) and heated under reflux for 4 h. The solvent was evaporated under reduced pressure, and the product was isolated by flash chromatography using ethyl acetate as an eluent. For pyridone **2e**, acetonitrile was used instead of toluene (reflux for 3 h).

*Ethyl 5-benzoyl-1-(2,2-dimethoxyethyl)-4-oxo-1,4-dihydropyridine-2-carboxylate* (**2a**). Yield 0.1067 g (90%), yellow crystals. IR (ATR) ν 3053, 2932, 2836, 1720, 1652, 1630, 1475, 1294, 829, 701. ^1^H NMR (400 MHz, CDCl_3_) *δ* 1.41 (t, *J* = 7.1 Hz, 3H, Me), 3.42 (s, 6H, 2MeO), 4.34 (d, *J* = 4.5 Hz, 2H, CH_2_N), 4.40 (q, *J* = 7.1 Hz, 2H, CH_2_), 4.53 (t, *J* = 4.5 Hz, 1H, CH), 7.07 (s, 1H, H-3 Py), 7.43 (t, *J* = 7.6 Hz, 2H, H-3, H-5 Ph), 7.55 (t, *J* = 8.0 Hz, 2H, H-4 Ph), 7.79 (s, 1H, H-6 Py), 7.86 (dd, *J* = 8.0 Hz, *J* = 1.4 Hz, 2H, H-2, H-6 Ph). ^13^C NMR (126 MHz, DMSO-*d*_6_) *δ* 13.7, 54.2, 55.4, 62.5, 102.7, 122.3, 127.9, 128.4 (2C), 129.1 (2C), 133.2, 136.9, 141.6, 147.2, 162.0, 174.4, 193.2. HRMS (ESI) m/z [M + H]^+^. Calculated for C_19_H_22_NO_6_: 360.1453. Found: 360.1447.

*Ethyl 1-(2,2-dimethoxyethyl)-5-(4-methoxybenzoyl)-4-oxo-1,4-dihydropyridine-2-carboxylate* (**2b**). Yield 0.0828 g (63%), light yellow oil. ^1^H NMR (500 MHz, CDCl_3_) *δ* 1.41 (t, *J* = 7.1 Hz, 3H, Me), 3.42 (s, 6H, 2MeO), 4.36 (d, *J* = 4.5 Hz, 2H, CH_2_N), 4.40 (q, *J* = 7.1 Hz, 2H, CH_2_), 4.53 (t, *J* = 4.5 Hz, 1H, CH), 7.06 (s, 1H, H-3 Py), 7.40 (d, *J* = 8.7 Hz, 2H, H-3, H-5 Ar), 7.79 (d, *J* = 8.7 Hz, 2H, H-2, H-6 Ar), 7.82 (s, 1H, H-6 Py). ^13^C NMR (151 MHz, CDCl_3_) *δ* 14.0, 55.5, 55.7, 55.9, 62.8, 103.3, 113.6, 124.8, 129.4, 129.8, 132.3, 140.2, 147.3, 162.4, 163.8, 175.7, 191.6. Anal. Calculated for C_20_H_23_NO_7_·0.5H_2_O: C 60.29; H 6.07; N 3.52. Found: C 59.98; H 6.20; N 3.47.

*Ethyl 5-(4-chlorobenzoyl)-1-(2,2-dimethoxyethyl)-4-oxo-1,4-dihydropyridine-2-carboxylate* (**2c**). Yield 0.0949 g (73%), brown viscous liquid. IR (ATR) ν 3033, 2978, 2862, 1729, 1660, 1630, 1481, 1246, 851, 767. ^1^H NMR (400 MHz, CDCl_3_) *δ* 1.41 (t, *J* = 7.1 Hz, 3H, Me), 3.42 (s, 6H, 2MeO), 4.36 (d, *J* = 4.5 Hz, 2H, CH*_2_*N), 4.40 (q, *J* = 7.1 Hz, 2H, CH_2_), 4.53 (t, *J* = 4.5 Hz, 1H, CH), 7.06 (s, 1H, H-3 Py), 7.40 (d, *J* = 8.7 Hz, 2H, H-3, H-5 Ar), 7.79 (d, *J* = 8.7 Hz, 2H, H-2, H-6 Ar), 7.82 (s, 1H, H-6 Py). ^13^C NMR (126 MHz, CDCl_3_) *δ* 14.0, 55.7, 55.8, 62.8, 103.2, 125.3, 128.2, 128.5 (2C), 131.0 (2C), 135.4, 139.4, 140.3, 148.1, 162.2, 175.6, 192.1. Anal. Calculated for C_19_H_20_ClNO_6_: C 57.95; H 5.12; N 3.56. Found: C 58.35; H 5.42; N 3.48.

*Ethyl 1-(2,2-dimethoxyethyl)-4-oxo-5-(thiophene-2-carbonyl)-1,4-dihydropyridine-2-carboxylate* (**2d**). Yield 0.0868 g (72%), dark yellow viscous liquid. IR (ATR) ν 3066, 2985, 2933, 2841, 1637, 1620, 1478, 1250, 849, 719. ^1^H NMR (400 MHz, CDCl_3_) *δ* 1.41 (t, *J* = 7.1 Hz, 3H, Me), 3.42 (s, 6H, 2MeO), 4.34 (d, *J* = 4.5 Hz, 1H, CH_2_N), 4.40 (q, *J* = 7.1 Hz, 2H, CH_2_), 4.53 (t, *J* = 4.5 Hz, 1H, CH), 7.09 (s, 1H, H-3 Py), 7.12 (dd, *J* = 4.9 Hz, *J* = 3.9 Hz, 1H, H-4 Th), 7.68 (dd, *J* = 4.9, *J* = 1.1 Hz, 1H, H-5 Th), 7.81 (s, 1H, H-6 Py), 7.85 (dd, *J* = 3.9, *J* = 1.1 Hz, 1H, H-3 Th). ^13^C NMR (126 MHz, CDCl_3_) *δ* 13.9, 55.7, 55.8, 62.8, 103.2, 125.1, 128.1, 128.8, 134.7, 135.1, 140.0, 143.7, 147.4, 162.2, 175.3, 184.2. Anal. Calculated for C_17_H_19_NO_5_S: C 55.88; H 5.24; N 3.83. Found: C 56.09; H 5.15; N 3.90.

*Ethyl 1-(2,2-dimethoxyethyl)-4-oxo-5-pivaloyl-1,4-dihydropyridine-2-carboxylate* (**2e**). Yield 0.0470 g (42%), brown viscous liquid. IR (ATR) ν 2960, 2837, 1731, 1628, 1480, 1247, 952, 805. ^1^H NMR (400 MHz, CDCl_3_) *δ* 1.28 (s, 9H, *t*-Bu), 1.40 (t, *J* = 7.1 Hz, 3H, Me), 3.41 (s, 6H, 2MeO), 4.26 (d, *J* = 4.7 Hz, 2H, CH_2_N), 4.38 (q, *J* = 7.1 Hz, 2H, CH_2_), 4.50 (t, *J* = 4.7 Hz, 1H, CH), 6.97 (s, 1H, H-3 Py), 7.41 (s, 1H, H-6). ^13^C NMR (126 MHz, CDCl_3_) *δ* 13.9, 26.2, 44.7, 55.6, 55.8, 62.7, 103.4, 123.9, 132.1, 139.7, 144.1, 162.4, 175.4, 209.8. HRMS (ESI) m/z [M + H]^+^. Calculated for C_17_H_26_NO_6_: 340.1766. Found: 340.1760.

*Diethyl 1-(2,2-dimethoxyethyl)-4-oxo-1,4-dihydropyridine-2,5-dicarboxylate* (**2f**). Yield 0.0324 g (30%), brown viscous liquid. IR (ATR) ν 2978, 2839, 1725, 1664, 1631, 1474, 1299, 867, 776. ^1^H NMR (400 MHz, CDCl_3_) *δ* 1.38 (t, *J* = 7.1 Hz, 3H, Me), 1.39 (t, *J* = 7.1 Hz, 3H, Me), 3.41 (s, 6H, 2MeO), 4.33 (d, *J* = 4.7 Hz, 2H, CH_2_N), 4.37 (q, *J* = 7.1 Hz, 2H, CH_2_), 4.38 (q, *J* = 7.1 Hz, 2H, CH_2_), 4.49 (t, *J* = 4.7 Hz, 1H, CH), 7.07 (s, 1H, H-3 Py), 8.18 (s, 1H, H-6 Py). HRMS (ESI) m/z [M + H]^+^. Calculated for C_15_H_22_NO_7_: 328.1406. Found: 328.1396. The spectral data are in accordance with the patent literature [11].

### 3.2. General Method for the Preparation of dihydropyrido[2,1-c][1,4]oxazine-1,8-diones **3**

1-(2,2-Dimethoxyethyl)-4-pyridone **2** (0.332 mmol) was stirred in hydrochloric acid (1: 1, 2 mL) for 24 h or, for **2b**, in concentrated hydrochloric acid (2 mL) for 3 h at room temperature and for 3 h under reflux. The precipitate formed was filtered. For compound **2f**, formic acid (85%, 2 mL) was used. The reaction mixture was stirred at room temperature for 3 h and heated at 85 °C for 4 h. After evaporation of the solvent, the product was isolated by flash chromatography using ethyl acetate as an eluent.

*7-Benzoyl-3-hydroxy-3,4-dihydropyrido[2,1-c][1,4]oxazine-1,8-dione* (**3a**). Yield 0.0506 g (53%), white powder, mp 187–188 °C. IR (ATR) ν 3228, 3078, 2489, 1665, 1637, 1470, 1213, 854, 748. ^1^H NMR (400 MHz, DMSO-*d*_6_) *δ* 4.29 (dd, *J* = 13.5 Hz, *J* = 3.3 Hz, 1H, C*H*H), 4.47 (dd, *J* = 13.5 Hz, *J* = 1.6 Hz, 1H, CH*H*), 6.08 (unresolved m, 1H, H-3), 6.99 (s, 1H, H-9), 7.50 (d, *J* = 7.7 Hz, 2H, H-3, H-5 Ph), 7.64 (t, *J* = 7.4 Hz, *J* = 1.0 Hz, 1H, H-4 Ph), 7.76 (dd, *J* = 8.4 Hz, *J* = 1.3 Hz, 2H, H-2, H-6 Ph), 8.28 (br.s, 1H, H-6), 8.72 (br.s, 1H, OH). ^13^C NMR (126 MHz, DMSO-*d*_6_) *δ* 52.4, 93.7, 121.1, 128.6 (2C), 129.3 (2C), 133.6, 136.5, 144.4, 158.1, 174.0, 192.9. HRMS (ESI) m/z [M + H]^+^. Calculated for C_15_H_12_NO_5_: 286.0714. Found: 286.0715.

*3-Hydroxy-7-(4-methoxybenzoyl)-3,4-dihydropyrido[2,1-c][1,4]oxazine-1,8-dione* (**3b**). Yield 0.0902 g (85%), beige powder, mp 205–206 °C. IR (ATR) ν 3060, 2934, 1684, 1599, 1268, 1153, 1021, 912, 843. ^1^H NMR (500 MHz, DMSO-*d*_6_) *δ* 3.84 (s, 3H, OMe), 4.25 (dd, *J* = 13.5 Hz, *J* = 3.6 Hz, 1H, C*H*H), 4.44 (dd, *J* = 13.5 Hz, *J* = 1.5 Hz, 1H, CH*H*), 6.07 (unresolved m, 1H, H-3), 6.94 (s, 1H, H-9), 7.03 (d, *J* = 8.9 Hz, 2H, H-3, H-5 Ar), 7.75 (d, *J* = 8.8 Hz, 2H, H-2, H-6 Ar), 8.18 (s, 1H, H-6), 8.66 (br.s, 1H, OH). ^13^C NMR (126 MHz, DMSO-*d*_6_) *δ* 52.1, 55.6, 93.6, 113.8 (2C), 121.2, 129.3, 129.9, 131.8 (2C), 135.9, 143.5, 158.2, 163.5, 174.2, 191.3. Anal. Calculated for C_16_H_13_NO_6_·0.25H_2_O: C 60.10; H 4.26; N 4.38. Found: C 60.17; H 3.98; 4.42.

*7-(4-Chlorobenzoyl)-3-hydroxy-3,4-dihydropyrido[2,1-c][1,4]oxazine-1,8-dione* (**3c**). Yield 0.0626 g (59%), white powder, mp 209–210 °C. IR (ATR) ν 3077, 1728, 1645, 1635, 1557, 1533, 1404, 1216, 1047, 732. ^1^H NMR (500 MHz, DMSO-*d*_6_) *δ* 4.32 (br.s, 1H, C*H*H), 4.45 (br.s, 1H, CH*H*), 6.10 (unresolved m, 1H, H-3), 6.90 (br.s, 1H, H-9), 7.57 (d, *J* = 8.9 Hz, 2H, H-3, H-5 Ar), 7.75 (d, *J* = 8.9 Hz, 2H, H-2, H-6 Ar), 8.28 (s, 1H, H-6), 8.67 (s, 1H, OH). ^13^C NMR (126 MHz, DMSO-*d*_6_) *δ* 52.0, 122.5, 128.6 (2C),128.7, 130.9 (2C), 135.5, 138.0, 144.6, 163.0, 174.8, 192.5 (2C were not observed). HRMS (ESI) m/z [M + H]^+^. Calculated for C_15_H_11_ClNO_5_: 320.0330. Found: 320.0326.

*3-Hydroxy-7-(thiophene-2-carbonyl)-3,4-dihydropyrido[2,1-c][1,4]oxazine-1,8-dione* (**3d**). Yield 0.0812 g (84%), beige powder, mp 206–207 °C. IR (ATR) ν 3071, 2904, 1730, 1638, 1407, 1209, 852, 749. ^1^H NMR (500 MHz, DMSO-*d*_6_) *δ* 4.26 (dd, *J* = 13.3 Hz, *J* =2.7 Hz, 1H, C*H*H), 4.46 (d, *J* = 13.3 Hz, 1H, CH*H*), 6.06 (s, 1H, H-3), 6.95 (s, 1H, H-9), 7.23 (dd, *J* = 4.9 Hz, *J* = 3.9 Hz, 1H, H-4 Th), 7.72 (dd, *J* = 3.9 Hz, *J* = 0.9 Hz, 1H, H-3 Th), 8.08 (dd, *J* = 4.9 Hz, *J* = 0.9 Hz, 1H, H-5 Th), 8.25 (s, 1H, H-6), 8.65 (d, *J* = 4.5 Hz, 1H, OH). ^13^C NMR (126 MHz, DMSO-*d*_6_) *δ* 51.8, 93.5, 122.1, 128.7, 129.3, 135.5, 135.7, 143.2, 143.3, 158.4, 174.4, 184.7 (1C was not observed). Anal. Calculated for C_13_H_9_NO_5_S: C 53.61; H 3.11; N 4.81. Found: C 53.64; H 3.39; N 4.90.

*3-Hydroxy-7-pivaloyl-3,4-dihydropyrido[2,1-c][1,4]oxazine-1,8-dione* (**3e**). Yield 0.0361 g (41%), brown liquid product. IR (ATR) ν 2972, 2935, 2909, 1697, 1479, 1364, 1216, 1060. ^1^H NMR (500 MHz, DMSO-*d*_6_) *δ* 1.18 (s, 9H, *t*-Bu), 4.33 (s, 2H, 4-CH_2_), 6.13 (br.s, 1H, H-3), 6.83 (s, 1H, H-9), 7.90 (s, 1H, H-6), 8.13 (s, 1H, OH). ^13^C NMR (126 MHz, DMSO-*d*_6_) *δ* 26.0, 44.0, 54.6, 88.4, 120.1, 131.9, 140.9, 159.7, 163.1, 174.9, 210.1. HRMS (ESI) m/z [M + H]^+^. Calculated for C_13_H_16_NO_5_: 266.1033. Found: 266.1028.

*Ethyl 3-hydroxy-1,8-dioxo-1,3,4,8-tetrahydropyrido[2,1-c][1,4]oxazine-7-carboxylate* (**3f**). Yield 0.0622 g (74%), brown powder, mp 193–194 °C. IR (ATR) ν 2985, 1734, 1695, 1575, 1450, 1306, 1200, 1112, 1012, 876, 797. ^1^H NMR (500 MHz, DMSO-*d*_6_) *δ* 1.26 (t, *J* = 7.1 Hz, 2H, Me), 4.21 (q, *J* = 7.1 Hz, 2H, CH_2_), 4.31 (br.s, 1H, CH), 4.41 (s, 1H, CH), 6.03 (s, 1H, H-3), 6.87 (s, 1H, H-9), 8.47 (s, 1H, H-6), 8.61 (s, 1H, OH). ^13^C NMR (126 MHz, DMSO-*d*_6_) *δ* 14.2, 52.0, 60.2, 93.6, 119.8, 123.1, 135.4, 146.5, 158.6, 163.8, 173.7. HRMS (ESI) m/z [M + H]^+^. Calculated for C_11_H_12_NO_6_: 254.0669. Found: 254.0665.

### 3.3. General Method for the Preparation of Compounds **4** and **5**

*Method A.* 3-Hydroxy-7-acyl-3,4-dihydropyrido[2,1-c][1,4]oxazine-1,8-dione **3** (0.317 mmol) was stirred with amine (0.381 mmol) and ammonium acetate (0.0245 g, 0.317 mmol) in a mixture of toluene (1 mL) and methanol (1 mL) at 90 °C of an oil bath for 7–12 h. The precipitate obtained was filtered and recrystallized in a mixture of toluene and ethanol.

*Method B.* 3-Hydroxy-7-acyl-3,4-dihydropyrido[2,1-c][1,4]oxazine-1,8-dione **3** (0.317 mmol) was stirred with amine (0.381 mmol) and acetic acid (22.8 mg, 0.380 mmol) in a mixture of toluene (1 mL) and methanol (1 mL) at 90 °C of an oil bath for 7–12 h. The precipitate obtained was filtered and recrystallized in a mixture of toluene and ethanol.

*9-(4-Methoxybenzoyl)-3,4,12,12a-tetrahydro-2H-pyrido[1’,2’:4,5]pyrazino[2,1-b][1,3]oxazine-6,8-dion* (**4a**). The reaction was carried out for 12 h. Method A. Yield 0.0775 g (69%), yellow powder, mp 222–223 °C. IR (ATR) ν 2956, 2875, 1673, 1640, 1573, 1467, 1384, 783. ^1^H NMR (400 MHz, DMSO-*d*_6_) *δ* 1.61 (dm, *J* = 13.2 Hz, 1H, H-3*’*), 1.64–1.84 (m, 1H, H-3), 3.22 (td, *J* = 12.9 Hz, *J* = 3.2 Hz, 1H, CH), 3.84 (s, 3H, OMe), 3.90 (td, *J* = 11.8 Hz, *J* = 2.7 Hz, 1H, CH), 4.07 (dd, *J* = 11.3 Hz, *J* = 4.7 Hz, 1H, CH), 4.28 (dd, *J* = 14.1 Hz, *J* = 3.5 Hz, 1H, CH), 4.46 (dd, *J* = 14.0 Hz, *J* = 4.2 Hz, 2H, CH), 5.28 (t, *J* = 3.6 Hz, 1H, H-12a), 6.94 (s, 1H, H-7), 7.02 (d, *J* = 8.9 Hz, 2H, H-3, H-5 Ar), 7.75 (d, *J* = 8.9 Hz, 2H, H-2, H-6 Ar), 8.11 (s, 1H, H-10). ^13^C NMR (126 MHz, DMSO-*d*_6_) *δ* 25.0, 43.0, 50.2, 55.5, 67.1, 81.5, 113.7 (2C), 120.1, 129.6, 129.7, 131.7, 137.9 (2C), 142.9, 157.0, 163.3, 175.0, 192.0. HRMS (ESI) m/z [M + H]^+^. Calculated for C_19_H_19_N_2_O_5_: 355.1288. Found: 355.1294.

*9-(4-Chlorobenzoyl)-3,4,12,12a-tetrahydro-2H-pyrido[1′,2′:4,5]pyrazino[2,1-b][1,3]oxazine-6,8-dione* (**4b**). Method A. The reaction was carried out for 12 h. 3-Aminopropan-1-ol (47.6 mg, 0.634 mmol) was used. Yield 0.0853 g (75%), yellow powder, mp 259–261 °C. IR (ATR) ν 3020, 2950, 1674, 1627, 1572, 1452, 1357, 789. ^1^H NMR (500 MHz, DMSO-*d*_6_) *δ* 1.61 (dm, *J* = 12.5 Hz, 1H, H-3*’*), 1.70–1.82 (m, 1H, H-3), 3.22 (td, *J* = 13.0 Hz, *J* = 3.2 Hz, 1H, CH), 3.90 (td, *J* = 12.0 Hz, *J* = 2.4 Hz, 1H, CH), 4.07 (dd, *J* = 11.4 Hz, *J* = 4.8 Hz, 1H, CH), 4.32 (dd, *J* = 14.1 Hz, *J* = 3.5 Hz, 1H, CH), 4.43–4.50 (m, 2H, CH), 5.29 (t, *J* = 3.6 Hz, 1H, H-12a), 6.96 (s, 1H, H-7), 7.55 (d, *J* = 8.5 Hz, 2H, H-3, H-5 Ar), 7.75 (d, *J* = 8.5 Hz, 2H, H-2, H-6 Ar), 8.22 (s, 1H, H-10). ^13^C NMR (126 MHz, DMSO-*d*_6_) *δ* 25.0, 43.0, 50.3, 67.1, 81.4, 120.7, 128.4, 128.5 (2C), 131.0 (2C), 135.6, 137.9, 138.1, 144.2, 156.9, 175.0, 192.7. HRMS (ESI) m/z [M + H]^+^. Calculated for C_18_H_15_ClN_2_O_4_: 359.0789. Found: 359.0799.

*9-(Thiophene-2-carbonyl)-3,4,12,12a-tetrahydro-2H-pyrido[1’,2’:4,5]pyrazino[2,1-b][1,3]oxazine-6,8-dione* (**4c**). Method B. The reaction was carried out for 8h. Yield 0.0545 g (52%), yellow powder, mp 210–211 °C. IR (ATR) ν 2953, 1634, 1581, 1468, 1373, 1047, 714. ^1^H NMR (500 MHz, DMSO-*d*_6_) *δ* 1.61 (dm, *J* = 13.5 Hz, 1H, H-3*’*), 1.70–1.82 (m, 1H, H-3), 3.22 (td, *J* = 12.8 Hz, *J* = 3.2 Hz, 1H, CH), 3.89 (td, *J* = 11.9 Hz, *J* = 2.3 Hz, 1H, CH), 4.05 (dd, *J* = 11.3 Hz, *J* = 4.7 Hz, 1H, CH), 4.28 (dd, *J* = 14.2 Hz, *J* = 3.2 Hz, 1H, CH), 4.43–4.50 (m, 2H, CH), 5.27 (t, *J* = 3.6 Hz, 1H, H-12a), 6.97 (s, 1H, H-7), 7.22 (dd, *J* = 4.9 Hz, *J* = 4.0 Hz, 1H, H-4 Th), 7.74 (dd, *J* = 4.0 Hz, *J* = 0.7 Hz, 1H, H-3 Th), 8.05 (dd, *J* = 4.9 Hz, *J* = 0.7 Hz, 1H, H-5 Th), 8.19 (s, 1H, H-10). ^13^C NMR (126 MHz, DMSO-*d*_6_) *δ* 25.0, 43.0, 52.0, 67.1, 81.5, 120.4, 128.6, 128.9, 135.6, 137.9, 143.1, 143.4, 156.9, 174.6, 184.9 (1C was not observed). HRMS (ESI) m/z [M + H]^+^. Calculated for C_16_H_15_N_2_O_4_S: 331.0745. Found: 331.0753.

*9-Benzoyl-3,4,12,12a-tetrahydro-2H-pyrido[1’,2’:4,5]pyrazino[2,1-b][1,3]oxazine-6,8-dione* (**4d**). Method B. The reaction was carried out for 10 h. Yield 0.0319 g (31%), yellow powder, mp 245–246 °C. ^1^H NMR (400 MHz, DMSO-*d*_6_) *δ* 1.61 (dm, *J* = 14.6 Hz, 1H, H-3*’*), 1.67–1.84 (m, 1H, H-3), 3.22 (td, *J* = 13.0 Hz, *J* = 3.7 Hz, 1H, CH), 3.90 (td, *J* = 11.3 Hz, *J* = 2.5 Hz, 1H, CH), 4.07 (dd, *J* = 11.3 Hz, *J* = 4.5 Hz, 1H, CH), 4.31 (dd, *J* = 14.5 Hz, *J* = 3.5 Hz, 1H, CH), 4.42–4.51 (m, 2H, CH), 5.29 (t, *J* = 3.5 Hz, 1H, H-12a), 6.95 (s, 1H, H-7), 7.49 (t, *J* = 7.8 Hz, 2H, H-3, H-5 Ph), 7.62 (t, *J* = 7.8 Hz, 1H, H-4 Ph), 7.75 (d, *J* = 8.0 Hz, 2H, H-2, H-6 Ph), 8.18 (s, 1H, H-10). ^13^C NMR (126 MHz, DMSO-*d*_6_) *δ* 25.0, 50.2, 67.1, 81.5, 120.4, 128.4 (2C), 129.0, 129.1 (2C), 133.1, 136.8, 138.1, 143.6, 156.9, 175.0, 193.8. HRMS (ESI) m/z [M + H]^+^. Calculated for C_18_H_17_N_2_O_4_: 325.1188. Found: 325.1177.

*8-(4-Methoxybenzoyl)-2,3,11,11a-tetrahydro-1H-imidazo[1,2-a]pyrido[1,2-d]pyrazine-5,7-dione* (**5a**). Method A. The reaction was carried out for 12 h. Yield 0.0904 g (84%), brown powder, mp 249–250 °C;. IR (ATR) ν 2996, 2837, 1650, 1629, 1577, 1454, 1250, 786. ^1^H NMR (500 MHz, DMSO-*d*_6_) *δ* 2.99 (dt, *J* = 11.5 Hz, *J* = 7.4 Hz, 1H, CH), 3.20 (ddd, *J* = 11.2 Hz, *J* = 7.0 Hz, *J* = 3.9 Hz, 1H, CH), 3.39 (ddd, *J* = 11.1 Hz, *J* = 7.2 Hz, *J* = 3.8 Hz, 1H, CH), 3.54 (dt, *J* = 11.2 Hz, *J* = 7.3 Hz, 1H, CH), 3.84 (s, 3H, OMe), 3.92 (t, *J* = 11.8 Hz, 1H, CH), 4.52 (dd, *J* = 12.4 Hz, *J* = 3.7 Hz, 1H, CH), 4.75 (dd, *J* = 11.1 Hz, *J* = 3.6 Hz, 1H, CH), 6.79 (s, 1H, H-6), 7.01 (d, *J* = 8.8 Hz, 2H, H-3, H-5 Ar), 7.75 (d, *J* = 8.8 Hz, 2H, H-2, H-6 Ar), 8.09 (s, 1H, H-9) (NH was not observed). ^13^C NMR (126 MHz, DMSO-*d*_6_) *δ* 44.3, 44.8, 53.0, 55.5, 69.2, 113.6 (2C), 119.3, 129.0, 129.7, 131.7 (2C), 139.3, 143.3, 154.9, 163.2, 175.2, 192.2. HRMS (ESI) m/z [M + H]^+^. Calculated for C_18_H_18_N_3_O_4_: 340.1285. Found: 340.1297.

*8-(4-Chlorobenzoyl)-2,3,11,11a-tetrahydro-1H-imidazo[1,2-a]pyrido[1,2-d]pyrazine-5,7-dione* (**5b**). The reaction was carried out for 12 h. Method A. Yield 0.0850 g (78%), beige powder, mp 264–265 °C. IR (ATR) ν 2901, 1657, 1630, 1562, 1456, 1251, 769. ^1^H NMR (500 MHz, DMSO-*d*_6_) *δ* 2.95–3.03 (m, 1H, CH), 3.21 (ddd, *J* = 11.5 Hz, *J* = 7.1 Hz, *J* = 3.8 Hz, 1H, CH), 3.40 (ddd, *J* = 11.2 Hz, *J* = 7.1 Hz, *J* = 3.9 Hz, 1H, CH), 3.54 (dt, *J* = 11.4 Hz, *J* = 7.3 Hz, 1H, CH), 3.94 (t, *J* = 11.8 Hz, 1H, CH), 4.57 (dd, *J* = 12.4 Hz, *J* = 3.7 Hz, 1H, CH), 4.76 (dd, *J* = 11.1 Hz, *J* = 3.3 Hz, 1H, CH), 6.80 (s, 1H, H-6), 7.56 (d, *J* = 8.5 Hz, 2H, H-2, H-6 Ar), 7.75 (d, *J* = 8.5 Hz, 2H, H-2, H-6 Ar), 8.22 (s, 1H, H-9) (NH was not observed). ^13^C NMR (126 MHz, DMSO-*d*_6_) *δ* 44.3, 44.8, 53.1, 69.2, 119.9, 127.6, 128.5 (2C), 131.0 (2C), 135.8, 137.8, 139.5, 144.6, 154.7, 175.2, 192.8. HRMS (ESI) m/z [M + H]^+^. Calculated for C_17_H_15_ClN_3_O_3_: 344.0813. Found: 344.0802.

*8-(Thiophene-2-carbonyl)-2,3,11,11a-tetrahydro-1H-imidazo[1,2-a]pyrido[1,2-d]pyrazine-5,7-dione* (**5c**). Method B. The reaction was carried out for 7h. Yield 0.0480 g (48%), beige powder, mp 267–268 °C. ^1^H NMR (500 MHz, DMSO-*d*_6_) *δ* 2.99 (dt, *J* = 11.5 Hz, *J* = 7.4 Hz, 1H, CH), 3.20 (ddd, *J* = 11.1 Hz, *J* = 6.9 Hz, *J* = 3.9 Hz, 1H, CH), 3.39 (ddd, *J* = 11.2 Hz, *J* = 7.2 Hz, *J* = 3.8 Hz, 1H, CH), 3.54 (dt, *J* = 11.2 Hz, *J* = 7.3 Hz, 1H, CH), 3.92 (t, *J* = 11.8 Hz, 1H, CH), 4.51 (dd, *J* = 12.4 Hz, *J* = 3.7 Hz, 1H, CH), 4.75 (dd, *J* = 11.0 Hz, *J* = 3.6 Hz, 1H, CH), 6.81 (s, 1H, H-6), 7.22 (dd, *J* = 5.1 Hz, *J* = 3.9 Hz, 1H, H-4 Th), 7.76 (dd, *J* = 3.9 Hz, *J* = 0.8 Hz, 1H, H-3 Th), 7.76 (dd, *J* = 5.1 Hz, *J* = 0.8 Hz, 1H, H-5 Th), 8.18 (s, 1H, H-9). ^13^C NMR (126 MHz, DMSO-*d*_6_) *δ* 44.3, 44.8, 53.1, 69.2, 119.5, 128.2, 128.6, 135.5, 135.6, 139.3, 143.5, 154.8, 174.8, 185.0 (1C was not observed). HRMS (ESI) m/z [M + H]^+^. Calculated for C_15_H_14_N_3_O_3_S: 316.0756. Found: 316.0761.

### 3.4. General Method for the Preparation of Compound **6**

3-Hydroxy-7-acyl-3,4-dihydropyrido[2,1-c][1,4]oxazine-1,8-dione **3** (0.317 mmol) was stirred with benzene-1,2-diamine (41.2 mg, 0.381 mmol) and ammonium acetate (0.0245 g, 0.317 mmol) in a mixture of toluene (1 mL) and methanol (1 mL) at room temperature for 12 h. Then, the reaction mixture was refluxed for 2 h. The precipitate obtained was filtered and washed with H_2_O and EtOH. The product was recrystallized in a mixture of toluene and ethanol.

*3-Benzoyl-6H-benzo[4,5]imidazo[1,2-a]pyrido[1,2-d]pyrazine-2,13-dione* (**6a**). Yield 0.0800 g (71%), beige powder, mp 252–253 °C. IR (ATR) ν 3037, 2808, 1685, 1443, 1211, 1114, 1027, 807. ^1^H NMR (500 MHz, DMSO-*d*_6_) *δ* 5.84 (s, 2H, CH_2_), 6.80 (s, 1H, H-1), 7.18 (dd, *J* = 6.0 Hz, *J* = 3.2 Hz, 2H, Ar), 7.50 (t, *J* = 7.7 Hz, 2H, H-3, H-5 Ph), 7.55 (dd, *J* = 6.0 Hz, *J* = 3.2 Hz, 2H, Ar), 7.63 (t, *J* = 7.4 Hz, 1H, H-4 Ph), 7.81 (d, *J* = 7.4 Hz, 2H, H-2, H-6 Ph), 8.33 (s, 1H, H-4). ^13^C NMR (126 MHz, DMSO-*d*_6_) *δ* 51.7, 114.9, 122.0 (2C), 122.6, 128.2, 128.3 (2C), 129.2 (2C), 133.0, 137.0, 138.2, 142.4, 147.0, 150.2, 168.1, 175.1, 193.5 (2C were not observed). HRMS (ESI) m/z [M + H]^+^. Calculated for C_21_H_14_N_3_O_3_: 356.1035. Found: 356.1048.

*3-(4-Methoxybenzoyl)-6a,7-dihydro-6H-benzo[4,5]imidazo[1,2-a]pyrido[1,2-d]pyrazine-2,13-dione* (**6b**). Yield 0.1112 g (91%), yellow powder, mp 234–235 °C. IR (ATR) ν 2970, 2751, 1632, 1556, 1466, 1329, 1214, 1028, 840. ^1^H NMR (500 MHz; DMSO-*d*_6_) *δ* 3.84 (s, 3H, OMe), 5.82 (s, 2H, CH_2_), 6.82 (s, 1H, H-1), 7.02 (d, *J* = 8.8 Hz, 2H, H-3, H-5 Ar), 7.18 (dd, *J* = 6.1 Hz, *J* = 3.1 Hz, 2H, Ar’), 7.54 (dd, *J* = 6.1 Hz, *J* = 3.1 Hz, Ar*’*), 7.80 (d, *J* = 8.8 Hz, 2H, H-2, H-6 Ar), 8.26 (s, 1H, H-4). ^13^C NMR (126 MHz, DMSO-*d*_6_) *δ* 51.5, 55.5, 113.6 (2C), 115.0, 121.9 (2C), 125.2, 128.1, 128.7, 128.8, 129.7, 131.7 (2C), 137.3, 138.3, 146.1, 150.2, 163.1, 163.2, 175.1, 191.8. HRMS (ESI) m/z [M + H]^+^. Calculated for C_22_H_16_N_3_O_4_: 386.1141. Found: 386.1141.

*3-(4-Chlorobenzoyl)-6H-benzo[4,5]imidazo[1,2-a]pyrido[1,2-d]pyrazine-2,13-dione* (**6c**). Yield 0.0408 g (33%), beige powder, mp 251–252 °C. IR (ATR) ν 2971, 2756, 1633, 1597, 1466, 1256, 779. ^1^H NMR (500 MHz, DMSO-*d*_6_) *δ* 5.84 (s, 2H, CH_2_), 6.82 (s, 1H, H-1), 7.18 (dd, *J* = 6.0 Hz, *J* = 3.2 Hz, 2H, Ar*’*), 7.54 (dd, *J* = 6.0 Hz, *J* = 3.2 Hz, 2H, Ar*’*), 7.57 (d, *J* = 8.5 Hz, 2H, H-3, H-5 Ar), 7.80 (d, *J* = 8.5 Hz, 2H, H-2, H-6 Ar), 8.37 (s, 1H, H-4). ^13^C NMR (126 MHz, DMSO-*d*_6_) δ 55.5, 113.5, 115.0, 122.1 (2C), 123.1, 125.2, 127.7, 128.5 (2C), 131.1 (2C), 135.8, 137.9, 138.0, 142.0, 147.9, 150.2, 163.0, 175.1, 192.5. Anal. Calculated for C_21_H_12_ClN_3_O_3_: C 64.71; H 3.10; N 10.78. Found: C 64.64; H 3.73; N 10.60.

*3-(Thiophene-2-carbonyl)-6H-benzo[4,5]imidazo[1,2-a]pyrido[1,2-d]pyrazine-2,13-dione* (**6d**). Yield 0.0481 g (42%), orange powder, mp 243–244 °C. IR (ATR) ν 2875, 2760, 1635, 1557, 1464, 1326, 1211, 1061, 878. ^1^H NMR (500 MHz, DMSO-*d*_6_) *δ* 5.81 (s, 2H, CH_2_), 6.81 (s, 1H, H-1), 7.18 (dd, *J* = 6.0 Hz, *J* = 3.2 Hz, 2H, Ar), 7.23 (dd, *J* = 4.9 Hz, *J* = 3.9 Hz, 1H, H-4 Th), 7.54 (dd, *J* = 6.0 Hz, *J* = 3.2 Hz, 2H, Ar), 7.84 (dd, *J* = 3.9 Hz, *J* = 1.1 Hz, 1H, H-3 Th), 8.05 (dd, *J* = 4.9 Hz, *J* = 1.1 Hz, 1H, H-5 Th), 8.35 (s, 1H, H-4). ^13^C NMR (126 MHz, DMSO-*d*_6_) *δ* 51.7, 115.0, 122.1 (2C), 122.8, 128.1, 128.6 (2C), 135.7 (2C), 138.0, 141.9, 143.6, 146.8, 150.3, 163.1, 174.7, 184.6 (1C was not observed). Anal. Calculated for C_19_H_11_N_3_O_3_S: C 63.15; H 3.07; N 11.63. Found: C 62.83; H 3.33; N 11.25.

## Data Availability

Data are contained within the article and Appendix A.

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
