# Peer review of "The Construction of Polycyclic Pyridones via Ring-Opening Transformations of 3-hydroxy-3,4-dihydropyrido[2,1-c][1,4]oxazine-1,8-diones"

_molecules, 2023, doi:10.3390/molecules28031285_

Round 1

Reviewer 1 Report

Manuscript presented by Viktoria V. Viktorova et al. shows a study about synthesis of  polycyclic pyridones. The topic is very important due to new simpler synthesis aspects of these compounds.

An already well written and prepared manuscript. Easy to read and follow. Some aspects should be improved. I recommend the article to publish but first the paper should be corrected. My decision – reconsider after minor revision. Comments to be considered, in order to further improve the manuscript quality:

(1)   Please define what Authors mean “These molecules CAN be considered as novel“ in abstract? Are structures of these molecules commonly know? If the synthesized compounds are new connections? If only the synthesis method is innovative. This fact should be clearly defined in the manuscript.

(2)   Figure 1 does not contribute much to the topic of the manuscript. Consider removing.

(3)   The motivation of undertaking the research is too weak. Improve this aspect it. Focus on scheme 1 to describe.

(4)   Avoid lumping a lot of references as in 11-18 and all other. Instead summarize the main contribution of each referenced paper in a separate sentence. Eg. Please expand the sentence “The chemistry of these biologically important compounds is being actively developed not only for medicinal chemistry [5,6], but also in industry [1–3,7–9]”.

(5)   Please add more specification like NMR inert standard, what technique was used for preparation of FTIR, kind of ESI, type of column used in HR-MS and ect.

(6)   In the introduction, information about the importance of heterocycles in modern organic chemistry should be added. It should be underline the role of  pyridones derivatives. Include the information about synthesis methods of this class of compounds is also welcome. These aspects were described recently in other MDPI publications and should be adding to manuscript: Agrochemicals, 2(1), 34-46 (2023) DOI: https://doi.org/10.3390/agrochemicals2010004; Molecules, 27, 8409 (2022) DOI: https://doi.org/10.3390/molecules27238409; Molecules, 26, 1364 (2021) DOI:  https://doi.org/10.3390/molecules26051364;

(7)   I did not find a conclusion of presented study. Please complete it obligatory.

(8)   In supporting information file please add also HR-MS spectra.

(9)   Superscripts and subscripts as well as commas and periods should be checked. Avoid extra spaces and enters. Correct in whole manuscript.

(10)                      The English correction is necessary.

Author Response

1) Please define what Authors mean “These molecules CAN be considered as novel“ in abstract? Are structures of these molecules commonly know? If the synthesized compounds are new connections? If only the synthesis method is innovative. This fact should be clearly defined in the manuscript.

The approach is novel for the construction of polycyclic 4-pyridones. Also, all compounds obtained in this work are new, except for compound 2f. The text has been modified in accordance with the recommendations of the reviewer. The introduction has been improved.

“Besides, substituted 3-hydroxy-3,4-dihydropyrido[2,1-c][1,4]oxazine-1,8-dione was detected as the result of hydrolysis of the carboethoxy and dimethylacetal groups as a by-product in the synthesis of dolutegravir [10] Such structure also can be suggested as an possible intermediate for this heterocyclization [1,10]. To the best of our knowledge, only unsubstituted 3-hydroxy-3,4-dihydropyrido[2,1-c][1,4]oxazine-1,8-dione was prepared from comanic acid in pure form and used for the transformation with (R)-3-aminobutan-1-ol [3,11].”

2) Figure 1 does not contribute much to the topic of the manuscript. Consider removing.

Figure 1 is needed to show the most important polycyclic pyridones and their structural similarity with the compounds obtained in the work. To emphasize the pyridone ring, it was highlighted in red.

3) The motivation of undertaking the research is too weak. Improve this aspect it. Focus on scheme 1 to describe.

The motivation has been improved.

“We decided to study 3-hydroxy-3,4-dihydropyrido[2,1-c][1,4]oxazine-1,8-diones in more detail in order to find new directions for the construction of polycyclic pyridones through morpholinone ring-opening reactions. These molecules bear the hidden aldehyde moiety, which can determine their high reactivity towards nucleophiles via the tautomeric equilibrium. This strategy based on the transformation with diamines can open access to new cyclic fused pyridones, which are of interest for further design of biologically active compounds.”

4) Avoid lumping a lot of references as in 11-18 and all other. Instead summarize the main contribution of each referenced paper in a separate sentence. Eg. Please expand the sentence “The chemistry of these biologically important compounds is being actively developed not only for medicinal chemistry [5,6], but also in industry [1–3,7–9]”.

Many references include similar results, so they have been grouped. The sentence has been revised. “The chemistry of these heterocycles is actively developed not only for design of biologically important compounds [7–9], but also for effective preparation in industry [1–3,10–15].”

5) Please add more specification like NMR inert standard, what technique was used for preparation of FTIR, kind of ESI, type of column used in HR-MS and ect.

Information about standards for NMR spectroscopy is present in the experimental part, as well as for the FTIR method. ESI includes positive ionization (M+H+). We used direct injection for recording HRMS spectra.

6) In the introduction, information about the importance of heterocycles in modern organic chemistry should be added. It should be underline the role of  pyridones derivatives. Include the information about synthesis methods of this class of compounds is also welcome. These aspects were described recently in other MDPI publications and should be adding to manuscript: Agrochemicals, 2(1), 34-46 (2023) DOI: https://doi.org/10.3390/agrochemicals2010004; Molecules, 27, 8409 (2022) DOI: https://doi.org/10.3390/molecules27238409; Molecules, 26, 1364 (2021) DOI:  https://doi.org/10.3390/molecules26051364;

The additional references regarding 4-pyridones have been added.

7) I did not find a conclusion of presented study. Please complete it obligatory.

The conclusion is present at the end of the discussion and was not highlighted as a separate section.

8) In supporting information file please add also HR-MS spectra.

The HRMS spectra have been added in the supporting materials.

9) Superscripts and subscripts as well as commas and periods should be checked. Avoid extra spaces and enters. Correct in whole manuscript.

The spaces in the experimental section have been avoided. Superscripts and subscripts as well as commas and periods have been checked.

10) The English correction is necessary.

The English has been improved.

Author Response

1) “Pg 1………ethoxy group were detected as by-products [7] or suggested as intermediates [1,8]. To the best of our knowledge, in patent [9] –reword unsubstituted 3-hydroxy-3,4-dihydropyrido[2,1- c][1,4]oxazine-1,8-dione was prepared from comanic acid in pure form and described [3]. Besides, there are data on the Ugi reaction of a pyridone-bearing aldoacid with isonitriles for the synthesis of various piperazinone-fused pyridones [18].”

The sentence has been modified in according with the recommendation of the reviewer.

 “Substituted 3-hydroxy-3,4-dihydropyrido[2,1-c][1,4]oxazine-1,8-dione was also detected as the result of hydrolysis of the carboethoxy and dimethylacetal groups as a by-product in the synthesis of dolutegravir [10]. Such structure also can be suggested as an possible intermediate for this heterocyclization [1,10].”

2) Pg 3 “The 1H NMR spectra of products 3 in DMSO-d6 demonstrates the existence of only

the lactol form. The spectral feature of the tautomer is the presence of a downfield signal of the OH group at δ 8.13–8.72 ppm and an ABX system of the morpholinone moiety. For pivaloyl-substituted compound 3f, a singlet of the methylene group and a strongly broad- ened singlet of the CH proton were observed probably due to the rapid interconversion between different forms. Suggest using NMR to detect and characterize the carboxylate of the ring-opened form of one of 3a-3f at elevated pH.”

We carried out the reaction of compound 2c with sodium carbonate and sodium hydrocarbonate in ethanol under reflux, the reaction did not proceed, so the presence of the ammonium salt fragment is important for the ring-opening process.

3) “Pg 4 and thienoyl-substituted compounds 3a,d reacted more effectively in the conditions of method B and

gave products 4c,d in 31–52% yields. For propane-1,3-diamine, the trans- formation did not give the

desired polycyclic products selectively., but instead gave………………………... Next, binucleophiles

bearing two carbon atoms in the linker was used for the heterocyclization.”

The sentence has been changed, and the result was described in more detail.

“When propane-1,3-diamine was used as a binucleophile, we were not able to isolate the desired polycyclic products in pure form directly. The precipitates that formed always contained starting diamine in significant amounts.”

4) “Pg 6 tization under the action of atmospheric oxygen (Scheme 4, Table 4). The intermediate C was not isolated in pure form but was detected as a by-product in all cases. Carrying out the reaction under argon did not allow the selective formation of compound C. This result can indicate that the oxidation additionally promotes the reaction leading to most stable product 6. To obtain compounds 6 in pure form directly, the reaction was carried out at room temperature for 12 h and subsequent reflux for 2 h. Under these conditions the aromatization proceeded completely and pyridones 6 bearing the benzimidazole fragment were isolated in 33–91% yields. Explanation for wide range of yields? Pivaloyl pyridone 2e Should be 3e?? led to degradation products”

The reaction turned out to be sensitive to the nature of the acyl moiety, which probably determined side reactions. The text has been changed in according with the recommendation of the reviewer.

“The reaction turned out to be sensitive to the nature of the acyl moiety, which probably determined occurrence of side reactions. Pivaloyl pyridone 3e led to degradation products, which did not bear the t-Bu group.”

5) Pg 7 “Ethyl 5-acyl-4-oxo-4H-pyran-2-carboxylate 1 (0.33 mmol) was added to a cooled solution of 2,2-

dimethoxyethanamine (0.038 g, 0.36 mmol) in toluene (1 mL). The resulting mixture was stirred for 20 min

at room temperature (the precipitation of a finely crystalline precipitate was observed) and heated under

reflux for 4 h. The solvent was evaporated……”

The sentence has been corrected in according with the recommendation of the reviewer.